# Effects of cyproheptadine on body weight gain in children with nonorganic failure to thrive in Taiwan: A hospital-based retrospective study

Yi-Chun Lin[1,2], Hung-Rong Yen[1,3], Fuu-Jen Tsai[3,4,5], Chung-Hsing Wang[5,6], Lung-Chang Chien[7], An-Chyi Chen[5]*, Ro-Ting Lin[2]*

1 Department of Chinese Medicine, China Medical University Hospital, Taichung, Taiwan, 2 College of Public Health, China Medical University, Taichung, Taiwan, 3 School of Chinese Medicine, China Medical University, Taichung, Taiwan, 4 Department of Medical Research, China Medical University Hospital, Taichung, Taiwan, 5 Department of Pediatrics, China Medical University Children's Hospital, Taichung, Taiwan, 6 School of Medicine, College of Medicine, China Medical University, Taichung, Taiwan, 7 Department of Epidemiology and Biostatistics, University of Nevada, Las Vegas, School of Public Health, Las Vegas, NV, United States of America

* roting@mail.cmu.edu.tw (RTL); d8427@mail.cmuh.org.tw (ACC)

**Data Availability Statement:** Data cannot be shared publicly because of routine clinical and laboratory data from electronic medical records at

## Abstract

Failure to thrive (FTT) impairs the expected normal physical growth of children. This study aimed to evaluate the effects of cyproheptadine hydrochloride on growth parameters in prepubertal children with FTT. The medical records of prepubertal children who were newly diagnosed with FTT at China Medical University Hospital between 2007 and 2016 were retrospectively examined. The patients were divided into two groups depending on whether they had (T-group) or had not (NT-group) received cyproheptadine hydrochloride (0.3 mg/kg daily) for at least 14 days. The mean length of the treatment period was 97.22 days (range: 14–532 days). Weight, height, and body mass index were adjusted for age using the median values in the growth charts for Taiwanese boys and girls as the reference. A total of 788 patients aged 3–11 years were enrolled, 50 in the T-group and 738 in the NT-group. No statistically significant difference in the median age-adjusted weight value was noted between the T-group and NT-group during the follow up period. In the T-group, age-adjusted weight and body mass index were inversely associated with age ($P < 0.001$, $P < 0.001$) and positively associated with medication duration ($P = 0.026$, $P = 0.04$). Our findings underscore the positive association between cyproheptadine hydrochloride treatment and weight gain among prepubertal children. Further prospective clinical studies with a. longer and consistent treatment course is warranted.

## Introduction

Children with growth retardation account for 5–10% of pediatric outpatient visits [1, 2]. Failure to thrive (FTT) describes a state of undernutrition among children who are unable to

the CMUH. Data are available from the CMUH Institutional Data Access (contact via website: https://www.cmuh.cmu.edu.tw/NewsInfo/ NewsArticle?no=3568 or email: bdc@mail.cmuh. org.tw) for researchers who meet the criteria for access to confidential data.

**Funding:** The study was supported by grants from the China Medical University Hospital (DMR-107-154). The funders had no role in study design, data collection and analysis, decision to publish, or preparation of the manuscript.

**Competing interests:** The authors have declared that no competing interests exist.

maintain the expected growth velocity according to age- and sex-specific growth charts [3]. FTT is associated with long-term negative health and developmental consequences, including short stature, cognitive disorders, behavioral abnormalities [4], and higher risk of respiratory syncytial virus infection and hospitalization [5]. Thus, it is immensely important for children to obtain sufficient nutrition for their growth and development from birth until the end of puberty [6].

FTT can be categorized into two types—namely, organic and nonorganic. Organic FTT describes children with poor growth and underlying medical conditions, whereas nonorganic FTT refers to children with poor growth but without underlying medical conditions [3]. More than 86% of FTT cases in hospitalized children have nonorganic etiologies, and this percentage is likely to be higher in the outpatient setting [3].

Treatment for nonorganic FTT focuses on behavioral and environmental modifications to stimulate the appetite of children and consequently increase their oral food intake [7]. Behavioral interventions include the provision of health education about correct nutritional concepts and appropriate feeding skills to children and their caregivers, whereas environmental modifications include the provision of suitable dining environments, atmospheres, and appliances to patients [8]. Behavioral and environmental modifications as well as food supplementation are applied as first-line therapy, albeit often in vain [8, 9]. Therefore, pharmacological stimulation is employed as adjunctive therapy [10, 11].

Studies have shown that histaminergic and serotonergic receptor systems modulate feeding behaviors [12–14]. Furthermore, findings from numerous studies support the use of appetite stimulants such as cyproheptadine hydrochloride (CH) for the promotion of weight gain in children with or without underlying medical conditions who exhibit poor growth [7, 11, 15–19]. Nonetheless, owing to procedural differences among the studies, it is difficult to make comparisons, draw conclusions, and establish clear guidelines for CH administration. For instance, the drug dosage, treatment duration, and age of the study subjects (prepubertal children and adolescents) differed in three previous randomized clinical trials of nonorganic FTT [11, 18, 19]. Additionally, children possess different feeding skills at different ages and exhibit different growth trends before versus during puberty [20–22]. Hence, we retrospectively investigated the effects of CH use on weight in prepubertal children with nonorganic FTT who were treated at a hospital in central Taiwan from 2007 to 2017.

## Materials and methods

### Study design

Data were collected from the records of children treated for nonorganic FTT at the pediatric endocrinology and gastroenterology outpatient clinic at China Medical University Hospital Children's Hospital Medical Center (CMUCHMC) in Taichung, Taiwan, between January 1, 2007 and December 31, 2017. CMUCHMC is not only a tertiary university-affiliated medical center but also a medical research and education institution in central Taiwan.

### Study subjects and variables

The electronic medical records of children diagnosed with FTT at CMUH between January 1, 2007 and December 31, 2017 were retrospectively reviewed. The collected data included date of birth, date of encounter, diagnosis, treatment, body height (BH), body weight (BW), puberty stage, medical prescriptions, maternal height, paternal height, birth weight, and gestational age at birth.

The inclusion criteria were as follows: (i) boys aged 3–11 years and girls aged 3–10 years at the first visit; (ii) a diagnosis satisfying the criteria of codes 783.41, 783.40, and 783.43 of the

International Classification of Disease, 9th Revision, Clinical Modification, as well as other FTT-related codes (as ancillary documents); and (iii) BW below the 3rd percentile or body mass index (BMI) below the 5th percentile (as derived from growth charts and matched for age and sex) at the first visit. Children <3 years-old at the first visit were excluded owing to the difficulty in distinguishing organic FTT from nonorganic FTT in this age group according to a recent report [23]. The exclusion criteria were as follows: (i) diagnosis of pathological FTT; (ii) diagnosis of a congenital or organic disease; (iii) long-term (>3 months) use of appetite stimulants other than CH including traditional Chinese medicines; (iv) intake of CH (0.3 mg/kg/day) for <14 days; (v) essential information missing from the electronic medical records (e.g., BW, BH, and puberty onset as indicated by breast, testicular, or public hair development, voice change, and menarche).

Because our study was retrospective, the allocation of treatment modalities was based on previous choices made by patients and their parents in consultation with their physicians. The data for patients who either did not receive CH or received CH at a dosage of 0.3 mg/kg/day for >14 days were retrieved, and BW, BH, and medication duration were assessed. The patient's height was measured in the hospital using a wall-mounted stadiometer. Maternal height and paternal height were recorded during outpatient follow-up. BMI was calculated as weight divided by height squared ($kg/m^2$).

## Statistical analysis

The heights, weights, and BMIs of the patients in our study were normalized for age using the standard values in the growth charts for Taiwanese children and adolescents as the reference. The growth charts are based on World Health Organization and health-related physical fitness standards [22]. Each value for each patient was matched to the median chart value for a child of the same age and sex as the patient. The patient value was then divided by the corresponding chart value, and the product was multiplied by 100. The adjusted values are referred to as % BH, %BW, and %BMI. Higher percentages indicate greater concurrence with standard values. Owing to unobtainable raw data, we were unable to calculate the standard deviation scores for height, weight, and BMI. The t-test was used to compare continuous variables between the control and treatment groups. The effects of medication duration on height, weight, and BMI were analyzed using linear mixed models with a first-order autoregressive structure, taking time, age, sex, and treatment group into account. All statistical analyses were performed using SAS version 9.4 software (SAS Institute, Cary, NC).

## Ethical considerations

This study was conducted in accordance with the principles outlined in the Declaration of Helsinki and was approved by the Research Ethics Committee of China Medical University and CMUH [CMUH106-REC3-069(CR-2)]. The retrospective analysis in this study was performed using routine clinical and laboratory data from electronic medical records at CMUH, and the data were anonymously accessed. Hence, the institutional review board waived the need for informed consent forms.

## Results

A total of 4,096 patients with a diagnosis related to FTT were initially enrolled. Subsequently, 2,914 patients treated with medications other than CH and 394 patients diagnosed with pathological FTT or a congenital or organic disease were excluded. Ultimately, 788 eligible patients were included in this study. They were divided into two groups: those who had received CH

treatment (0.3 mg/kg/day, the T-group, $N = 50$) and those who had not (the NT-group, $N = 738$) (Fig 1).

## Characteristics of patients with FTT

The number of outpatient clinic visits was higher in the T-group than in the NT-group. Most patients (38%) in the T-group had visited the clinic >5 times compared with none in the NT-group (Table 1). More than half (57.9%) of the patients the NT-group had visited the clinic only twice.

The demographic and anthropometric characteristics of the study groups at the first visit are summarized in Table 2. The mean age, height, and weight for all patients at diagnosis were 7.3 ± 2.4 years, 111.0 ± 13.5 cm, and 17.7 ± 4.5 kg, respectively. Additionally, the mean BMI, maternal height, and paternal height were 14.2 ± 1.1, 155.4 ± 4.8 cm, and 168.0 ± 5.5 cm, respectively. Mean age, height, weight, and BMI were significantly lower in the T-group than in the NT-group ($P = 0.0032$, 0.0218, 0.0023, and 0.0067, respectively). The differences in height, weight, and BMI between the two groups were the result of the difference in mean age. Thus, we normalized heights, weights, and BMIs for age using the sex-matched median values new growth charts for Taiwanese children and adolescents as the reference. After adjustment, no differences in %BH, %BW, or %BMI were observed between the T-group and NT-groups (Table 2).

Table 3 presents the modeling results for the effects of selected determinants on health outcomes in all subjects with FTT. The median %BW value did not differ significantly between the T-group and NT-group. Nonetheless, as the medication duration increased, median %BW and %BMI increased in a linear manner; both were inversely associated with age ($P < 0.001$, $P < 0.001$) and positively associated with medication duration ($P = 0.026$, $P = 0.04$).

Table 4 presents the modeling results for the effects of selected determinants on health outcomes in patients with FTT who received CH treatment. In this group, as in all patients, median %BW and %BMI were inversely associated with age ($P < 0.001$, $P < 0.001$) and positively and linearly associated with medication duration ($P < 0.001$, $P = 0.006$). Each additional day of CH treatment increased the median %BW value by 0.121 units. This is equivalent to a weight gain of 26.45 g/day, which was calculated by multiplying the daily weight gain unit (0.121) by the average standard weight of the T-group (21.86 kg). The following formula was used to determine the average standard weight:

$$\frac{21.7 \times 362 \; girls + 22.0 \times 426 \; boys}{788 \; total}$$

Additionally, extending the medication duration increased the median %BMI value by 0.095 units per day. Nevertheless, we were unable to assess whether the linear relationship between weight gain velocity and medication duration could extend beyond 4 months. It was because the medication duration in our study varied widely, ranging from 14 days to 532 days, and the number of subjects was insufficient for stratified analysis.

## Discussion

### Main findings

Our findings reveal a positive association between CH intake and weight gain among prepubertal children. We found that continuous intake of CH (0.3 mg/kg/day) for at least 14 days (mean: 3.24 months) increased the weight gain velocity and BMI in a linear fashion among prepubertal children (age range: 3–11 years) with mild-to-moderate wasting (60–74% of the median weight standard) but without medical conditions.

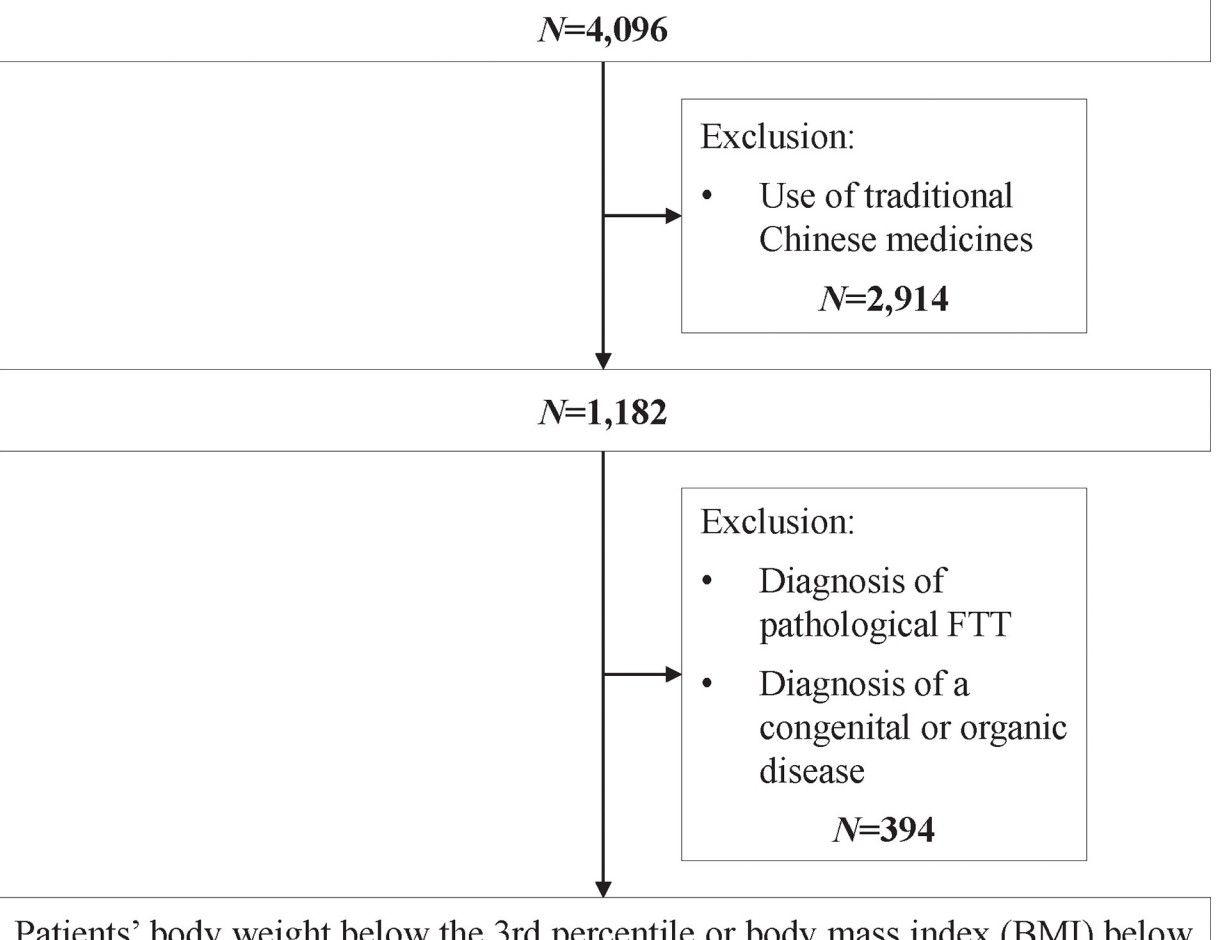

Patients diagnosed with diseases satisfying the criteria of codes 783.41, 783.40, and 783.43 of the International Classification of Disease, 9th Revision, Clinical Modification, as well as other failure to thrive (FTT) related codes (as ancillary documents) from 2007 to 2017 at China Medical University Hospital.

*N*=4,096

Exclusion:
- Use of traditional Chinese medicines

*N*=2,914

*N*=1,182

Exclusion:
- Diagnosis of pathological FTT
- Diagnosis of a congenital or organic disease

*N*=394

Patients' body weight below the 3rd percentile or body mass index (BMI) below the 5th percentile (as derived from growth charts and matched for age and sex) at the first visit.

*N*=788

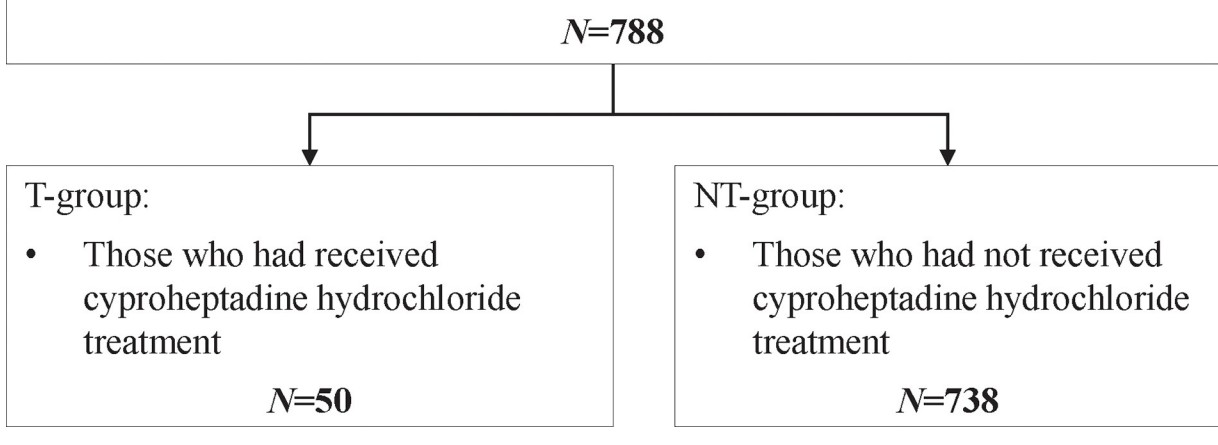

T-group:
- Those who had received cyproheptadine hydrochloride treatment

*N*=50

NT-group:
- Those who had not received cyproheptadine hydrochloride treatment

*N*=738

**Fig 1. Flow chart.**

**Table 1. Visiting status of subjects with failure to thrive.**

| Number of visits | NT-group | T-group | ALL |
|---|---|---|---|
| | N (%) | N (%) | N (%) |
| 1 | 62 (8.4) | 0 (0) | 62 (7.9) |
| 2 | 427 (57.9) | 13 (26) | 440 (55.8) |
| 3 | 183 (24.8) | 5 (10) | 188 (23.9) |
| 4 | 66 (8.9) | 13 (26) | 79 (10.0) |
| ≥5 | 0 (0) | 19 (38) | 19 (2.4) |

## Possible explanation

Low doses of CH (0.1–0.3 mg/kg/day) have been shown to have a pronounced effect on weight gain in children (S1 Table). Increasing the CH dose from 0.1 to 0.3 mg/kg/day increased weight gain velocity from 11.66 to 22.32 g/day and accelerated the attainment of normal weight in children in random control trials. The CH dosage for prepubertal children with non-organic FTT was the same in our study and that by Rerksuppaphol et al. [19]; however,

**Table 2. Comparison of the demographic and anthropometric characteristics of the failure to thrive groups at the first visit.**

| Variable | ALL | NT-group | T-group | P value |
|---|---|---|---|---|
| | N = 788 | N = 738 | N = 50 | |
| **All patients** | | | | |
| Age, year | 7.3 ± 2.4 | 7.4 ± 2.4 | 6.3 ± 2.5 | 0.0032** |
| Height, cm | 111.0 ± 13.5 | 111.3 ± 13.4 | 106.8 ± 14.3 | 0.0218** |
| Weight, kg | 17.7 ± 4.5 | 17.8 ± 4.5 | 15.9 ± 4.1 | 0.0023** |
| BMI | 14.2 ± 1.1 | 14.2 ± 1.1 | 13.8 ± 1.1 | 0.0067** |
| **Standard value for girls (N = 362)[a]** | | | | |
| Height, cm | 119.8 ± 12.8 | 120.0 ± 12.6 | 115.6 ± 16.0 | 0.1578 |
| Weight, kg | 23.2 ± 5.7 | 23.3 ± 5.7 | 21.7 ± 6.9 | 0.2498 |
| BMI | 15.9 ± 0.6 | 15.9 ± 0.6 | 15.8 ± 0.7 | 0.4865 |
| **Standard value for boys (N = 426)[a]** | | | | |
| Height, cm | 123.3 ± 14.2 | 123.9 ± 14.1 | 116.4 ± 14.2 | 0.0040**** |
| Weight, kg | 25.3 ± 6.9 | 25.6 ± 6.9 | 22.0 ± 6.4 | 0.0051** |
| BMI | 16.3 ± 0.8 | 16.3 ± 0.8 | 15.9 ± 0.7 | 0.0066** |
| **Age-adjusted value for all patients[b]** | | | | |
| Height, % | 91.2 ± 3.5 | 91.1 ± 3.5 | 91.9 ± 3.1 | 0.1275 |
| Weight, % | 73.2 ± 5.8 | 73.2 ± 5.7 | 73.3 ± 6.7 | 0.8616 |
| BMI, % | 88.3 ± 7.3 | 88.3 ± 7.3 | 87.0 ± 8.1 | 0.2091 |
| **Paternal height, cm** | 168.0 ± 5.5 | 168.0 ± 5.5 | 168.1 ± 5.3 | 0.9569 |
| **Maternal height, cm** | 155.4 ± 4.8 | 155.4 ± 4.8 | 155.8 ± 4.7 | 0.5892 |
| **Mid-parental height[c], cm** | 162.5 ± 7.6 | 162.4 ± 7.6 | 164.0 ± 7.0 | 0.1616 |

All data are expressed as mean ± standard deviation. The t-test was used to compare the continuous variables between the control and treatment groups.

[a]The standard values for height, weight, and BMI were derived from the growth charts for Taiwanese boys and girls at ages corresponding to those of the patients. The chart values at the 50th percentile was used.

[b]The percentages for height, weight, and BMI were calculated as follows: patient value ÷ standard value × 100.

[c]Calculated the mid-parental height by adding 6.5 cm to the average of both parents' heights for boys or subtracting 6.5 cm from the average of both parents' heights for girls.

**P <0.05.

BMI, body mass index.

Table 3. Estimated effects of determinants on health outcomes in all subjects (*N* = 788).

| Determinant | %BH | | %BW | | %BMI | |
|---|---|---|---|---|---|---|
| | Estimate | *P* value | Estimate | *P* value | Estimate | *P* value |
| Intercept | 89.955 | <0.001** | 78.126 | <0.001** | 96.569 | <0.001** |
| Time | 0.000 | 0.141 | 0.001 | <0.001** | 0.001 | 0.004** |
| Treatment[a] vs. control[b] | 0.719 | 0.719 | -0.333 | 0.686 | -1.863 | 0.047** |
| Male vs. female | 0.495 | 0.043** | 1.096 | 0.009 ** | 0.507 | 0.291 |
| Age | 0.102 | 0.044** | -0.783 | <0.001** | -1.150 | <0.001** |
| Medication duration | 0.016 | 0.261 | 0.089 | 0.026** | 0.088 | 0.040** |

[a]Treatment: cyproheptadine hydrochloride (0.3 mg/kg/day, >14 days).

[b]Control: no treatment.

**$P < 0.05$.

BH, body height; BW, body weight; BMI, body mass index.

the mean daily weight gain in our study was higher (26.45 g/day vs. 22.32 g/day) (S1 Fig). The results of Nemati et al. [24] suggest that CH loses effectiveness at higher doses: in mice fed 5, 10, or 20 mg/kg of CH per day, weight and food intake increased, did not change, or decreased, respectively. The exact mechanism underlying the effects of low-dose CH on weight gain in children is not entirely understood; hence, future studies on this matter are warranted.

The present study revealed a positive linear relationship between median %BW during treatment and medication duration. In a previous randomized controlled trial, CH use led to rapid and early weight gain in children with growth deficiencies, particularly in the first 4 months of treatment [25]. According to a chart review study on CH for stimulant-induced weight loss, the median time to a response is 65 days [26]. Considerable weight gain occurred within 2 weeks in the study by Rerksuppaphol et al. [19], but required 1–4 months in the study by Sant'Anna et al. [7]. Mahachoklertwattana et al. [11] showed that CH therapy increased weight within 2 months and that weight gain velocity proportionately declined as the length of the treatment increased. In our study, the mean CH duration was 97.22 days (3.24 months), which echoes the findings of the abovementioned studies on children with poor growth. Nevertheless, the length of the treatment in our study varied widely, ranging from at least 14 days to 532 days. We did not have a sufficient number of case subjects for a stratified analysis to determine whether the linear relationship between weight gain velocity and medication duration extended beyond 4 months. Clinical trials are required to explore this matter.

Table 4. Estimated effects of determinants on health outcomes in the T-group (*N* = 50).

| Determinant | %BH | | %BW | | %BMI | |
|---|---|---|---|---|---|---|
| | Estimate | *P* value | Estimate | *P* value | Estimate | *P* value |
| Intercept | 91.614 | <0.001** | 81.851 | <0.001** | 97.588 | <0.001** |
| Time | -0.001 | 0.292 | 0.00005 | 0.974 | 0.002 | 0.269 |
| Male vs. female | 0.164 | 0.857 | 1.263 | 0.511 | 1.275 | 0.462 |
| Age | 0.024 | 0.889 | -1.359 | <0.001** | -1.675 | <0.001** |
| Medication duration | 0.022 | 0.017** | 0.121 | <0.001** | 0.095 | 0.006** |

The T-group received cyproheptadine hydrochloride (0.3 mg/kg/day, >14days).

**$P < 0.05$.

BH, body height; BW, body weight; BMI, body mass index.

As summarized in a retrospective review, the side effects of CH are fairly benign in children with feeding problems [7]. Reported side effects such as tachycardia, constipation, diarrhea, irritability, and sleepiness are resolved by decreasing the dosage of CH or discontinuing the treatment [7, 27, 28]. Because side effects directly affect CH dosage and treatment duration, they also indirectly affect weight gain. Future research exploring the potential intermediary role of side effects on the association between CH treatment and body weight is suggested.

## Possible mechanism

Although unresolved, CH most likely promotes weight gain by acting as an appetite stimulant, thereby increasing calorie intake and causing weight gain. Two hypotheses have been proposed to explain this phenomenon. The first hypothesis suggests that CH increases appetite by directly activating the hypothalamic appetite center [19, 29]. This has been shown in animal models and has clinical relevance as the effects of CH on 5-HT2 and H1 receptors [12, 14, 30, 31] are also observed in humans [15, 32, 33]. Additionally, Comer et al. [34] showed that CH increases the number of eating occasions, suggesting that its actions may be more related to postprandial satiety/hunger signaling than to a food reward pathway.

The second hypothesis is based on the effects of CH on gastric activity [27, 28]. Previous studies using animal models revealed that CH decreases gastric tone and increases gastric compliance via 5-HT2A and 2B receptors, resulting in gastric fundic relaxation [35, 36]. Others found that CH barricade of serotonin receptors in animal intestines diminished the secretory effects of serotonin, thereby increasing feeding volume and calorie intake [37, 38]. It has been suggested that CH therapy increases growth velocity in underweight children by increasing the serum level of insulin-like growth factor-I (IGF-I) [11]. However, Razzaghy-Azar et al. [39] dispute this contention: in their study, the IGF-1 level was only slightly elevated and still below the normal range after CH therapy. Thus, further molecular studies of additional pathways are warranted.

## Study strengths and limitations

CMUCHMC, the site of our study, is not only one of the first National Children's Hospitals recognized by the Ministry of Health and Welfare in Taiwan but also the largest children's hospital in central Taiwan. It receives a large number of visits (e.g., 179,832 clinic visits in 2016) and is very representative of the demography in central Taiwan. Therefore, the strengths of this study included its large sample size and real-world use of CH. Additionally, the longitudinal dataset allowed us to assess the effects of CH over time in a diverse group of patients. Lastly, the weight gain (26.45 g/day) achieved in our study using 0.3 mg/kg/day was greater than that reported in previous studies, as was the speed with which normal weight was attained.

The present study also has some limitations. First, it had a retrospective study design that involved a chart review; hence, the causal relationship between CH treatment and weight gain cannot be confirmed. Second, the electronic medical records used for this study did not include information on socioeconomic status (e.g., household income) or confounding factors (e.g., disease activity, use of other drugs in other hospitals or clinics during the observation period, health education assessment). A previous study showed that a low household income could limit access to adequate diets, particularly for older children, and indicated that parents and caregivers might require dietary guidance to ensure sufficient quantity and quality of the foods eaten at home and to foster healthy eating habits among children [40]. Owing to the lack of information on household income in the present study, we might have underestimated the benefits of CH treatment. Although we did not consider prior behavioral or environmental interventions in the present study, we note that health education, which includes information

on behavioral and environmental modification, is the first-line treatment for children with non-organic FTT at our hospital. Hence, variations in health behaviors among the study participants should be small. However, previous studies have shown that health education often fails to reach its expected goal [8, 9].

Third, it was difficult to accurately exclude subjects with feeding disorders or functional gastrointestinal disorders via a retrospective review of medical records only. Pediatric feeding disorders are common: their reported percentages in children with and without delayed development are 25% and 80%, respectively. The consequences of feeding disorders such as growth failure can be severe [41]. Because severity potentially affects weight gain velocity, we might have underestimated the benefits of CH treatment.

Fourth, the children in the T-group visited our hospital and hence may have received more medical attention than did those with fewer visits. To compensate, we incorporated the number of visits into the time variable of our linear model, as shown in Table 4. Finally, because most of the patients in our study, which was conducted in central Taiwan, were Han Chinese, its generalizability to other ethnic groups may be limited.

## Conclusion

The use of CH (0.3 mg/kg/day, >14 days) is beneficial for prepubertal children with non-organic FTT in terms of promoting weight gain and quick attainment of normal weight.

## Supporting information

**S1 Table. Cyproheptadine use in underweight and/or malnourished children.**
(PDF)

**S1 Fig. Mean weight gain velocity in underweight and/or malnourished children with cyproheptadine hydrochloride.**
(PDF)

## Author Contributions

**Conceptualization:** An-Chyi Chen, Ro-Ting Lin.

**Data curation:** Yi-Chun Lin, Chung-Hsing Wang.

**Formal analysis:** Yi-Chun Lin, Fuu-Jen Tsai, Lung-Chang Chien, Ro-Ting Lin.

**Funding acquisition:** Yi-Chun Lin, Hung-Rong Yen.

**Investigation:** Yi-Chun Lin, Chung-Hsing Wang.

**Methodology:** Lung-Chang Chien, Ro-Ting Lin.

**Project administration:** Yi-Chun Lin.

**Resources:** Yi-Chun Lin, Hung-Rong Yen, Fuu-Jen Tsai, Chung-Hsing Wang, An-Chyi Chen, Ro-Ting Lin.

**Software:** Lung-Chang Chien, Ro-Ting Lin.

**Supervision:** An-Chyi Chen, Ro-Ting Lin.

**Writing – original draft:** Yi-Chun Lin.

**Writing – review & editing:** Yi-Chun Lin, An-Chyi Chen, Ro-Ting Lin.

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
