## [Decision Letter · Decision Letter 0]

6 May 2021

PONE-D-21-05718

Effects of cyproheptadine on body weight gain in children with nonorganic failure to thrive in Taiwan: a hospital-based retrospective study

PLOS ONE

Dear Dr. Lin,

Thank you for submitting your manuscript to PLOS ONE. After careful consideration, we feel that it has merit but does not fully meet PLOS ONE’s publication criteria as it currently stands. Therefore, we invite you to submit a revised version of the manuscript that addresses the points raised during the review process.

We look forward to receiving your revised manuscript.

Kind regards,

Tai-Heng Chen, M.D.

Academic Editor

PLOS ONE

1 Please ensure that your manuscript meets PLOS ONE's style requirements, including those for file naming. The PLOS ONE style templates can be found at

2. To comply with PLOS ONE submission guidelines, in your Methods section, please provide additional information regarding your statistical analyses. For more information on PLOS ONE's expectations for statistical reporting, please see https://journals.plos.org/plosone/s/submission-guidelines.#loc-statistical-reporting.

Reviewers' comments:

Reviewer's Responses to Questions

**Comments to the Author**

1. Is the manuscript technically sound, and do the data support the conclusions?

Reviewer #1: Yes

Reviewer #2: Yes

Reviewer #3: Partly

2. Has the statistical analysis been performed appropriately and rigorously? 

Reviewer #1: No

Reviewer #2: Yes

Reviewer #3: I Don't Know

3. Have the authors made all data underlying the findings in their manuscript fully available?

Reviewer #1: Yes

Reviewer #2: No

Reviewer #3: Yes

4. Is the manuscript presented in an intelligible fashion and written in standard English?

Reviewer #1: Yes

Reviewer #2: Yes

Reviewer #3: Yes

5. Review Comments to the Author

Reviewer #1: Lin, et al. submit a manuscript reporting a retrospective review of 788 children seen in the China Medical University Hospital for failure to thrive between 2007 and 2016. Fifty children received cyproheptadine in an attempt to improve weight gain, while the remaining 738 were untreated. The investigators compared measures of height, weight, and body mass index at baseline. Standardized measures adjusting for age showed that there was no difference between the groups at baseline. Based on modeling, the investigators identified a linear relationship between the duration of cyproheptadine treatment and measures of weight and BMI.

Specific comments:

1. Line 37: It is unclear what the authors mean by “%BH, %BW, and %BMI (50th percentile for age and sex)” It does not appear to mean percentiles for height, weight, and BMI. Does it mean the percentage of the 50th percentile for these measures? If so, this is an unusual way to express the data. The authors should analyze their data in terms of percentiles or SD scores.

2. Lines 39-40: It is unclear whether the authors are referring to measures obtained at baseline or at follow up.

3. Lines 45 and 202: What is meant by “we determined by consensus that CH was helpful…”

4. Line 59: Recommend “…children hospitalized with FTT have nonorganic etiologies…”

5. Lines 68-70 and 70-72: These two passages are redundant

6. Various places in the manuscript: Reference is made to “prepuberty children.” This should be “prepubertal children.”

7. Various places in the manuscript: Cyproheptadine is abbreviated to CH, but this is not done consistently.

8. Lines 142-145: The fact that the majority of patients in the NT group only had two visits to the clinic while the T group visited 5 or more times is a confounder because the increased medical attention and focus on weight may have altered feeding practices.

9. Table 2: What is “standard height/weight/BMI?”

10. Line 203: I think the authors mean 0.3 mg/kg daily.

11. In addition to the unusual expression of relative body measures, the authors do not discuss the changes in body weight in individual patients. The best approach would be to compare the change in height, weight, and BMI percentiles or SD scores in the T group vs. the NT group. It is not clear why they did not do this.

12. It is not clear what this study adds to an already fairly extensive literature.

13. Additional data would be informative, including the dose used, duration of treatment, and pubertal status.

Reviewer #2: The manuscript by Yi-Chun Lin and colleagues describes a retrospective cohort study looking at the effect of cyproheptadine on growth parameters in pre-pubertal children with non-organic failure to thrive. They showed that the group treated with cyproheptadine had statistically significant association with medication duration and %BW and %BMI. They were able to include a large number of patients compared to prior studies, but were limited due to its retrospective nature and multiple possible confounders.

Major comments/suggestions/questions:

Were the investigators able to gather information about prior behavioral or environmental interventions as well as nutritional supplements? Since these are common first line treatments it would be interesting to see if these patients had previously or concurrently used these treatments.

Since there have been many randomized control trials, it should be emphasized how this single center retrospective review adds to the known body of literature. One aspect that could be emphasized is the larger sample size. Another could be the real-world use of cyprohepatidine.

Table 1 – The title could be made more clear with listing the number of visits as opposed to "visiting status" and visit 1, etc

Z scores for weight and height and BMI for age should be used and compared between groups. This is the standard to assess nutritional status in children. There is not much utility in comparing weights and heights without adjusting for age and sex. Also, it would great to include mid-parental height and accounted for it if possible.

Table 3 and 4 are confusing and require more explanation in the results section. There appears to be a negative association between treatment vs control on % BMI in table 3 based on how it is displayed. Could these results be better shown with a graph?

The results section mentions an “equivalent weight gain of 26.43g/day,” but it is unclear how that number was obtained. Please explain how that was calculated.

It would be beneficial to include the dose and duration of cyproheptadine use in the methods section or as part of the results.

Were side effects reported and could they be included in this manuscript? Regardless of that data being available, there should also be a discussion of potential side effects of using cyproheptadine in the discussion section.

In the discussion it is mentioned that the higher the CH dose the child received the more weight velocity was attained, but the comparison between this real-world study and the RCTs can not lead to this conclusion. It might be inferred from the comparison but unless a full study is done, the conclusion the authors draw can not be made. This should be changed in the discussion.

Table 5 in general seems unnecessary and could be moved to being a supplemental table.

Minor comments:

De-identified data should be made available upon request

The increased number of clinic visits seen between the treatment groups should also be included as a possible confounder. The more frequent visits might indicate that the parents are more likely to be motivated to try an intervention and use other therapies.

Reviewer #3: This retrospective study examines the effects of cyproheptadine on weight gain and vertical growth in a small cohort of 50 children ages 3-11 diagnosed with nonorganic failure to thrive treated with cyproheptadine between 2007-2017 in an academic tertiary care center in Taiwan. These children are compared to a fairly large untreated cohort. Using “Linear Mixed Modeling”, the authors conclude that cyproheptadine improved %BMI and %BW in children with nonorganic failure to thrive. I think this has potential to support published data on the benefits of cyproheptadine for this specific age group.

Major Critiques:

1. This study does not add significant new contributions to the body of literature, as numerous published articles (including a systematic review and two prospective trials appropriately cited in the text) highlight the beneficial effects of cyproheptadine on growth. The authors cite differences in dosing and population standardization in prior studies as reasons to publish this data, but the cohort has significant age variation, and the authors do not clearly delineate if/what variation in cyproheptadine dosing was used in the patients in their cohort. Of note, suggestions for dosing exist for cyproheptadine as an appetite stimulant in medication dosing reference resources such as UpToDate.

Minor Critiques:

- Few minor grammatical errors should be addressed throughout the manuscript

PAGE 8, line 104: Recommend defining what qualifies as a “long-term medicines”. Would this be other appetite stimulants? Stimulant medications for ADHD that might suppress appetite? Would a multivitamin or other medication such as polyethylene glycol that would most likely be unrelated to growth exclude a patient?

PAGE 9, lines 117-119: Recommend including what specific statistical tests were used to compare treatment groups (ie t-test is cited in Table 2). Additional description of how “Linear Mixed Models” were used for statistical analysis is extremely important, as the conclusions are completely based on this analysis.

PAGE 13, Table 2: Very “busy” table. Could simplify to facilitate interpretation by excluding “N” columns and instead include the “N” data below column or row headings, and combine the Mean SD columns into Mean (SD)

PAGE 14, Table 3: This table does not clearly communicate desire results. Interpreting results in this format is not intuitive. Unclear what “Intercept” and “Time” refer to. Better represented graphically?

PAGE 15, Table 4: Similarly, a confusing table. Unclear what “Intercept” and “Time” refer to. Better represented graphically?

PAGE 15, Line 202-206: It is unclear where the outcomes stated in the main findings (0.3 mg of CH x14 days) is associated with increased BW gain and BMI are represented in the figures/tables.

PAGE 16, Line 211-213: Discussion about dose of cyproheptadine for appetite stimulation and subsequent reference to Table 5 is disjointed. Strongly recommend rewording.

PAGE 16, Table 5: The dosing of cyproheptadine should be introduced in the “Methods” section. The first mention of the dose of cyproheptadine occurs in the “Discussion section”

Figure 1: This figure is not cited in the text. The figure shows patients treated with traditional Chinese medicine were excluded, but this is not mentioned in the body of the text as one of the exclusion criteria

6. PLOS authors have the option to publish the peer review history of their article (what does this mean?). If published, this will include your full peer review and any attached files.

Reviewer #1: No

Reviewer #2: No

Reviewer #3: No

---

## [Author Response · Author response to Decision Letter 0]

29 Jun 2021

Dear Editors and Reviewers,

We wish to resubmit our manuscript titled “Effects of cyproheptadine on body weight gain in children with nonorganic failure to thrive in Taiwan: a hospital-based retrospective study” (PONE-D-21-05718R1) to PLOS One for further consideration. 

We thank the reviewers for their in-depth reading of our manuscript and their insightful comments. We have carefully considered their remarks and have modified our manuscript accordingly. As a result, we believe that our manuscript is much improved and hopefully now acceptable for publication in your journal.

Our point by point responses to the reviewers’ comments and questions are below.

Sincerely yours,

Ro-Ting Lin

Associate Professor of College of Public Health, China Medical University, Taiwan

---

## [Decision Letter · Decision Letter 1]

28 Jul 2021

PONE-D-21-05718R1

Effects of cyproheptadine on body weight gain in children with nonorganic failure to thrive in Taiwan: a hospital-based retrospective study

PLOS ONE

Dear Dr. Lin,

Thank you for submitting your manuscript to PLOS ONE. After careful consideration, we feel that it has merit but does not fully meet PLOS ONE’s publication criteria as it currently stands. Therefore, we invite you to submit a revised version of the manuscript that addresses the points raised during the review process.

We look forward to receiving your revised manuscript.

Kind regards,

Tai-Heng Chen, M.D.

Academic Editor

PLOS ONE

Journal Requirements:

Reviewers' comments:

Reviewer's Responses to Questions

**Comments to the Author**

1. If the authors have adequately addressed your comments raised in a previous round of review and you feel that this manuscript is now acceptable for publication, you may indicate that here to bypass the “Comments to the Author” section, enter your conflict of interest statement in the “Confidential to Editor” section, and submit your "Accept" recommendation.

Reviewer #1: (No Response)

Reviewer #2: (No Response)

2. Is the manuscript technically sound, and do the data support the conclusions?

Reviewer #1: Yes

Reviewer #2: Yes

3. Has the statistical analysis been performed appropriately and rigorously? 

Reviewer #1: Yes

Reviewer #2: Yes

4. Have the authors made all data underlying the findings in their manuscript fully available?

Reviewer #1: No

Reviewer #2: No

5. Is the manuscript presented in an intelligible fashion and written in standard English?

Reviewer #1: Yes

Reviewer #2: Yes

6. Review Comments to the Author

Reviewer #1: 1. Table 4 indicates that medication duration is positively associated with gains in %BW and %BMI. Could there be a selection bias here that explains this relationship? Those who had a good response might be more likely to continue medication use, while those who did not respond might abandon the treatment early.

2. Lines 211-213: The inability to assess the relationship beyond 4 months of treatment should be stated in the Results section as well.

Reviewer #2: In this revised version they have responded to many of the comments from reviewers but the following are comments that should still be addressed prior to acceptance of the manuscript:

Major comments/suggestions:

It would be more useful to have a mid-parental height as opposed to seeing the maternal and paternal heights separately

Please include units for variables in Table 2. For example in rows for standard value for girls/boys should include (cm, kg) and for age-adjusted values should include (%). It is described in footnote, but should also be included in the table.

Tables 3 and 4 are difficult to interpret alone without the explanation found in the text. Please display this information graphically in addition to showing the tables. This will make the data easier to interpret and will be visually more impactful.

Minor comments:

Line 99 is missing closed parenthesis

Line 101 should say “choices made by patients and their parents in consultation with their physicians”.

The formula for the calculation of average standard weight should be included in the methods and not the results section.

It would be nice if the new figure comparing this study with the others was included as a supplemental figure and referenced in the discussion section. It seems a shame for this figure to be only for the reviewers.

7. PLOS authors have the option to publish the peer review history of their article (what does this mean?). If published, this will include your full peer review and any attached files.

Reviewer #1: No

Reviewer #2: No

---

## [Author Response · Author response to Decision Letter 1]

12 Aug 2021

Dear Reviewers,

We would like to thank you for reviewing our revised manuscript and giving us insightful comments. We have carefully reviewed your suggestions, responded to your comments, and revised our manuscript. As a result, we believe that our revised manuscript is much improved and hopefully now acceptable. We have attached a point-by-point response file in response to your comments for your review. We look forward to hearing from you soon.

Sincerely yours,

Ro-Ting Lin

Associate Professor of College of Public Health, China Medical University, Taiwan

---

## [Decision Letter · Decision Letter 2]

5 Oct 2021

Effects of cyproheptadine on body weight gain in children with nonorganic failure to thrive in Taiwan: a hospital-based retrospective study

PONE-D-21-05718R2

Dear Dr. Lin,

We’re pleased to inform you that your manuscript has been judged scientifically suitable for publication and will be formally accepted for publication once it meets all outstanding technical requirements.

Kind regards,

Tai-Heng Chen, M.D.

Academic Editor

PLOS ONE

Reviewers' comments:

Reviewer's Responses to Questions

**Comments to the Author**

1. If the authors have adequately addressed your comments raised in a previous round of review and you feel that this manuscript is now acceptable for publication, you may indicate that here to bypass the “Comments to the Author” section, enter your conflict of interest statement in the “Confidential to Editor” section, and submit your "Accept" recommendation.

Reviewer #1: All comments have been addressed

Reviewer #2: All comments have been addressed

2. Is the manuscript technically sound, and do the data support the conclusions?

Reviewer #1: Yes

Reviewer #2: Yes

3. Has the statistical analysis been performed appropriately and rigorously? 

Reviewer #1: Yes

Reviewer #2: Yes

4. Have the authors made all data underlying the findings in their manuscript fully available?

Reviewer #1: Yes

Reviewer #2: No

5. Is the manuscript presented in an intelligible fashion and written in standard English?

Reviewer #1: Yes

Reviewer #2: Yes

6. Review Comments to the Author

Reviewer #1: (No Response)

Reviewer #2: The manuscript by Yi-Chun Lin and colleagues describes a retrospective cohort study looking at the effect of cyproheptadine on growth parameters in pre-pubertal children with non-organic failure to thrive. They showed that the group treated with cyproheptadine had statistically significant association with medication duration and %BW and %BMI. They were able to include a large number of patients compared to prior studies, but were limited due to its retrospective nature and multiple possible confounders.

In this revised version they have responded to all of the comments from reviewers. The analysis is done in an appropriate manner and the data is presented in a clear manner.

7. PLOS authors have the option to publish the peer review history of their article (what does this mean?). If published, this will include your full peer review and any attached files.

Reviewer #1: No

Reviewer #2: No

---

## [Editor Report · Acceptance letter]

11 Oct 2021

PONE-D-21-05718R2 

Effects of cyproheptadine on body weight gain in children with nonorganic failure to thrive in Taiwan: a hospital-based retrospective study 

Dear Dr. Lin:

I'm pleased to inform you that your manuscript has been deemed suitable for publication in PLOS ONE. Congratulations! Your manuscript is now with our production department. 

Kind regards, 

on behalf of

Dr. Tai-Heng Chen 

Academic Editor

PLOS ONE